# Train, Freeze or Exit: Dynamic Module-wise Fine-Tuning under Memory Constraints

## Abstract

Full fine-tuning delivers strong performance on large language models (LLMs), but its high memory cost limits practical adoption. Although numerous methods have been proposed to reduce memory usage, they rarely model memory cost explicitly and cannot make dynamic adjustments. These limitations hinder fine-tuning performance under memory constraints and complicate the selection of suitable configurations. To address this, we propose Three-State Module Scheduling (TriMS), a dynamic fine-tuning framework that assigns each module in the model to one of three states (trainable, frozen, or early exit). With this three-state formulation, TriMS can quantitatively estimate memory usage while clearly characterizing training configurations. During fine-tuning, TriMS constructs a performance–cost estimator, which is continuously updated by monitoring activation gradients and resource consumption, to evaluate candidate actions (e.g., shrinking or expanding trainable modules). By selecting actions with the best benefit–cost trade-off, TriMS achieves efficient fine-tuning under strict memory budgets. Extensive experiments across diverse tasks and models demonstrate that TriMS effectively performs dynamic module scheduling under memory constraints. At moderate resource limit (i.e., 80% of the peak memory required for full fine-tuning), TriMS matches or even outperforms the best baselines, consistently ranking among the top two methods. More importantly, under stricter constraints, where existing approaches often fail to adapt, TriMS maintains strong performance (e.g., achieving accuracy within 1.5% of full fine-tuning at just 60% of the memory cost).

## 1 Introduction

In recent years, Large Language Models (LLMs) have demonstrated impressive advancements in the field of natural language processing (NLP), enabling a wide range of applications (Du et al., 2023; Hadi et al., 2023). Fine-tuning pre-trained LLMs has become the standard paradigm for adapting them to specific domains or tasks, often yielding strong performance. However, full fine-tuning introduces substantial challenges, most notably in terms of GPU memory and training efficiency (Kaddour et al., 2023; Li et al., 2024). To reduce peak memory cost, researchers have proposed various techniques, including gradient checkpointing (Chen et al., 2016), Parameter-Efficient Fine-Tuning (PEFT) (Han et al., 2024), quantization (Zhao et al., 2024), zeroth-order optimizers (Chen et al., 2024), and model pruning (Ma et al., 2023).

Despite these advances, existing approaches still face notable limitations when fine-tuning under strict resource constraints. First, these methods **rarely model memory cost explicitly**. Most memory-efficient methods reduce memory usage without relying on quantitative or controllable modeling, often requiring trial runs to find feasible configurations. Some dynamic approaches (Liu et al., 2024b; Devoto et al., 2024) further analyze memory during training, but still depend on empirical observations rather than explicit predictive models. As a result, resource utilization remains hard to control, complicating deployment and tuning in practice. Second, they also **lack dynamic adaptability.** Current methods typically rely on static parameter allocations (e.g., a fixed LoRA rank), whereas fine-tuning demands are inherently dynamic, varying across layers and training stages. For example, lower layers may require more adaptation early on, while higher layers become critical later—a stage-dependent pattern confirmed in prior studies (Liu et al., 2024b; Peng et al., 2025). A

static allocation is insufficient to capture such evolving requirements, limiting both resource utilization and final performance.

To address these limitations, we propose a dynamic fine-tuning framework called **Thr**ee-State **M**odule **S**cheduling (**TriMS**). During training, TriMS assigns each module (e.g., a transformer block) to one of three states—trainable, frozen, or early exit. Such a three-state formulation not only defines the training configuration but also enables quantitative estimation of memory and time costs. Furthermore, TriMS incorporates a contiguity constraint that requires trainable modules to form a continuous interval, which transforms dynamic adjustment into the selection among seven candidate actions. During fine-tuning, TriMS monitors activation gradients and resource usage to construct and update a performance–cost estimator. Based on the evaluations from the estimator, it then selects the action that offers the best trade-off. Through this mechanism, TriMS enables dynamic module scheduling under strict memory budgets, allowing for efficient fine-tuning with limited resources. Extensive experiments across multiple models and tasks show that TriMS efficiently performs dynamic module scheduling under certain memory constraints. To summarize, this paper makes the following major contributions.

- **Three-state module formulation.** We represent each module as trainable, frozen, or early exit, providing a unified state representation that enables explicit estimation of memory and time costs.

- **Dynamic module-wise fine-tuning method.** Building on the three-state formulation, we design TriMS, which incorporates a contiguity constraint and transforms dynamic adjustment into the selection among seven candidate actions. During training, TriMS monitors activation gradients and resource usage, updates a performance–cost estimator, and selects actions with the best trade-off, thereby realizing efficient module scheduling under strict memory budgets.

- **Extensive empirical validation.** We conduct comprehensive experiments across diverse tasks and model architectures, showing that TriMS consistently matches or outperforms strong baselines. In addition, under strict constraints, our method remains effective (e.g., achieving accuracy within 1.5% of full fine-tuning at 60% memory).

## 2 RELATED WORKS

In this section, we review related work on memory-efficient fine-tuning and dynamic adaptation strategies, and further analyze their limitations that motivate the design of our TriMS.

**Memory-efficient fine-tuning.** With the growing adoption of large pre-trained models, efficient fine-tuning under limited resources has become a key challenge. Existing techniques can be broadly categorized into three classes. First, parameter-efficient fine-tuning (PEFT) reduces costs by updating only a subset of parameters or inserting lightweight modules. Early work froze most layers and trained only the output (Donahue et al., 2014; Yosinski et al., 2014), while adapter-based methods introduced trainable modules into frozen layers (Houlsby et al., 2019; Lin et al., 2020). LoRA (Hu et al., 2021) extended this idea with low-rank adaptation, and QLoRA (Dettmers et al., 2023) applied it to quantized models, yielding large memory savings. Second, model compression reduces model size via quantization (Frantar et al., 2023; Zhao et al., 2024) or pruning (Ma et al., 2023). Third, training process optimization reduces peak memory usage during backpropagation, for example, with gradient checkpointing (Chen et al., 2016) or zeroth-order optimizers (Chen et al., 2024).

**Dynamic adaptation strategies.** Beyond memory-efficient methods, many studies show that fine-tuning often requires dynamic adjustments across training stages. Early strategies mainly tuned learning rates or gradually unfroze layers, e.g., gradual unfreezing (Howard & Ruder, 2018), Freeze-Out (Brock et al., 2017), and layer-wise learning rate decay (Mosbach et al., 2021). More recently, dynamic mechanisms have been integrated into PEFT, allowing flexible allocation of trainable parameters under constraints. Examples include AFLoRA (Liu et al., 2024b), which freezes LoRA modules based on gradients, ALaST (Devoto et al., 2024), which adaptively selects layers in vision models, and DLFT (Peng et al., 2025), which leverages sensitivity analysis during training. These approaches underscore the value of dynamic configurations, but most remain heuristic and lack a unified predictive framework, making resource utilization difficult to control.

**Limitations of existing methods.** Despite recent advances, existing approaches still exhibit notable limitations under memory constraints. First, they rarely model memory cost explicitly. Most

methods focus on how to reduce memory cost, yet they lack unified and predictive models of resource consumption. As a result, practitioners often rely on heuristic trial-and-error to find feasible configurations. Second, they lack dynamic adaptability. Most methods adopt static configurations (e.g., fixed LoRA ranks) that cannot capture the evolving demands of different layers and training stages (Howard & Ruder, 2018; Brock et al., 2017; Mosbach et al., 2021). Even recent dynamic approaches (Liu et al., 2024b; Devoto et al., 2024; Peng et al., 2025) remain largely heuristic, adjusting parameters without predictive guidance, and often fail under stricter budgets. These limitations highlight the need for a new framework that combines explicit resource modeling with dynamic and controllable scheduling.

## 3 PRELIMINARIES & PROBLEM FORMULATION

In this section, we introduce the foundations of our approach. We first describe the modular structure of large models and present a three-state encoding to uniformly represent different training configurations. We then provide memory and time estimation based on this encoding, enabling quantitative analysis of resource consumption. Finally, we formalize the dynamic module scheduling problem under memory constraints, which serves as the basis for our method.

### 3.1 MODULAR STRUCTURE AND THREE-STATE ENCODING

Large models can generally be viewed as a stack of modular components, such as the Transformer blocks in LLaMA and GPT. This modular structure provides a natural granularity for dynamic configuration during fine-tuning, enabling training configurations to be represented at the module level. To formalize this, we introduce a three-state encoding that assigns each module one of the following states:

- **Trainable (1)**: the module is updated during training, requiring both forward and backward propagation as well as additional storage for gradients and optimizer states;

- **Frozen (0)**: the module remains fixed, performing forward computation without the need for gradients or optimizer states to update parameters;

- **Early Exit (-1)**: training terminates before this module; the output from preceding modules is directly routed to the task head (e.g., the classification layer). This module and all subsequent ones are excluded from forward and backward computation, and no gradients or optimizer states are stored for them.

With this encoding, a model with $n$ modules $\{M_j\}_{j=1}^n$ can be represented as a state vector $\boldsymbol{s} = [s_1, \ldots, s_n] \in \{1, 0, -1\}^n$, where each entry specifies the state of a module. Figure 1 illustrates the encoding examples for a three-module model. For instance, the encoding $(1, 1, 1)$ corresponds to full fine-tuning, where all modules are trainable.

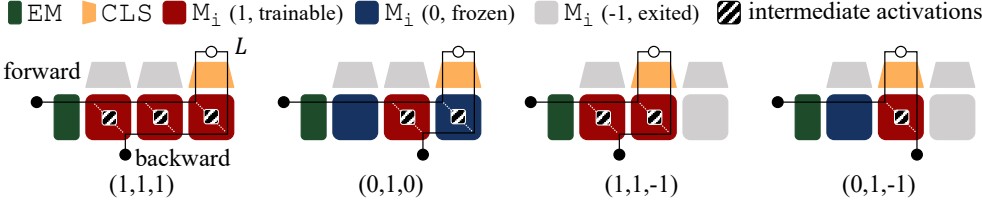

Figure 1: Examples of three-state encoding in a three-module model, showing full fine-tuning, partial freezing, and early exit configurations.

### 3.2 RESOURCE ESTIMATION WITH STATE ENCODING

Given the three-state encoding, we can explicitly model the resource consumption of different training configurations in terms of memory and time.

- **Memory.** For memory, the cost of a single module $\mathtt{M}^{(j)}$ consists of four parts: parameters $M_P^{(j)}$, gradients $M_G^{(j)}$, optimizer states $M_O^{(j)}$, and activations $M_A^{(j)}$. Here, $M_P$ depends on the number of parameters and storage precision; $M_O$ depends on the optimizer type (e.g., Adam requires two momentum terms per parameter); and $M_A$ mainly depends on batch size, sequence length, and model architecture, and can be reduced with activation checkpointing. Given a state vector $s$, we define

$$e = \min(\{j|s_j = -1\} \cup \{n\}) - 1, l = \min\{j|s_j = 1\}, s_{\text{train}} = \{j|s_j = 1\},$$

which denote the position of the last non-exit module, the position of the first trainable module, and the index of trainable modules, respectively. The total memory consumption can then be written as

$$M(s) = M_B + \sum_{j=1}^{n} M_P^{(j)} + \sum_{j=l}^{e} M_A^{(j)} + \sum_{j \in s_{\text{train}}} (M_G^{(j)} + M_O^{(j)}), \tag{1}$$

where $M_B$ represents fixed costs outside the modules. When exit modules are offloaded to the CPU, the parameter term changes from $\sum_{j=1}^{n} M_P^{(j)}$ to $\sum_{j=1}^{e} M_P^{(j)}$.

- **Time.** As for time, the cost of a single module $\mathtt{M}^{(j)}$ is decomposed into forward propagation $T_{FP}^{(j)}$, gradient propagation $T_{GP}^{(j)}$, and parameter update $T_{UP}^{(j)}$. Given state $s$, the forward cost is proportional to $e$, the backward cost depends on the interval $[l, e]$, and updates are only applied to trainable modules. Thus, the total time consumption can be approximated as

$$T(s) = \sum_{j=1}^{e} T_{FP}^{(j)} + \sum_{j=l}^{e} T_{GP}^{(j)} + \sum_{j \in s_{\text{train}}} T_{UP}^{(j)} + T_B, \tag{2}$$

where $T_B$ accounts for fixed costs such as embeddings and the output layer. This formula provides an approximate estimation and does not consider hardware-level optimizations.

### 3.3 PROBLEM DEFINITION

Based on the three-state encoding and resource estimation, we formulate dynamic module scheduling as a constrained optimization problem. The primary goal is to learn a scheduling policy $\mathcal{S}$ that maximizes performance under a strict memory budget, while training time serves as an auxiliary metric to compare the efficiency of feasible solutions.

Formally, a schedule is defined as $\mathcal{S} = [s_1, s_2, \ldots, s_T]$, where $s_t$ denotes the module state configuration at step $t$. The optimization problem is defined as finding the schedule $\mathcal{S}^*$ that

$$\mathcal{S}^* = \arg\max_{\mathcal{S}} V(\mathcal{S}) \quad \text{s.t.} \quad M(\mathbf{s}_t) \leq M_{\max}, \quad \forall \mathbf{s}_t \in \mathcal{S}, \tag{3}$$

where $V(\cdot)$ denotes the validation performance (e.g., accuracy or F1), and $M_{\max}$ is the maximum memory budget.

## 4 OUR METHOD: THREE-STATE MODULE SCHEDULING (TRIMS)

To enable resource-controllable dynamic module scheduling, we propose the TriMS framework. Building on the three-state encoding introduced in the previous section, we incorporate a contiguity constraint that requires all trainable modules to form a continuous block. This constraint reformulates dynamic adjustment as the selection among seven candidate actions, thereby simplifying the scheduling space while ensuring controllability. In this section, we first provide an overview of TriMS, then describe its design in detail, including the imposed constraints and action space, the construction and optimization of performance–cost estimators, the evaluation and filtering of candidate actions, and finally a brief discussion.

Figure 2 illustrates the overall workflow of TriMS. At each selection period ($\tau$ iterations), the model is trained under the current state $s_i$, producing observations including performance $L_i$, resource usage (memory $m_i$ and time $t_i$), and module activation gradients $g_i$ (step ①). These observations are used to continuously update two key estimators: the resource estimator, which predicts and calibrates memory $\hat{M}(s)$ and time usage $\hat{T}(s)$, and the reward estimator $\hat{V}(s, a)$, which models

the potential benefit of actions $\alpha \in \mathcal{A}$ (step ②). Candidate actions are then filtered according to structural and resource constraints (step ③), evaluated using both estimators, and scored to select the best action (step ④), which is applied to update the module state for training of the next selection step (step ⑤).

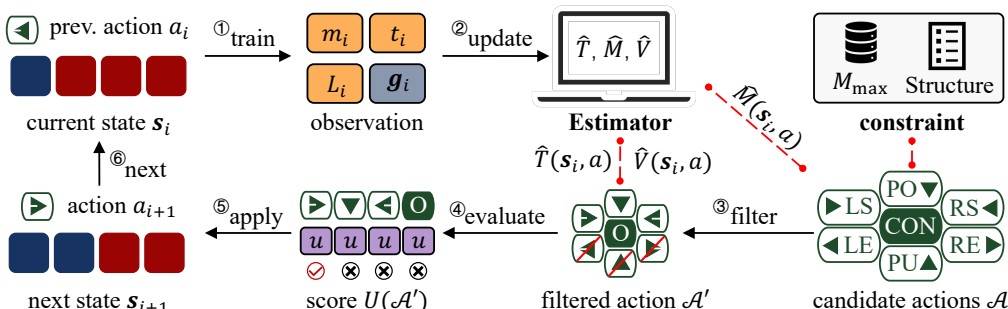

Figure 2: Overview of the TriMS framework.

## 4.1 STATE CONSTRAINTS & ACTION SPACE

Building on the three-state encoding, TriMS further introduces a **continuity constraint**, requiring all trainable modules to be arranged consecutively. This constraint helps the training in each selection period focus on a specific portion of the model and, from a computational and implementation perspective, greatly simplifies resource modeling and action operations. Under this constraint, a training state $s$ can be represented as a tuple $(l, r, e, n)$, where $l$ and $r$ denote the start and end positions of the trainable block, $e$ is the last non-exit module, and $n$ is the total number of modules.

Under this definition, the problem of dynamic module scheduling is reduced to the selection of a finite set of actions $\mathcal{A}$. The specific actions and their corresponding state transitions are summarized in Table 1. Notably, LS, RS, and PO reduce memory usage and time cost by shrinking the trainable or active range, while LE, RE, and PU increase the training scope and resource consumption. In addition, CON maintains the current configuration without changes. Through these seven actions, TriMS is able to flexibly and efficiently explore different training configurations while ensuring both structural continuity and resource controllability.

Table 1: Action space $\mathcal{A}$ under continuity constraint.

| Notation | Action | Transition | Description | Res. |
|---|---|---|---|---|
| LS | Left Shrink | $l \rightarrow l+1$ | Freeze the first trainable module | ↓ |
| LE | Left Expand | $l \rightarrow l-1$ | Unfreeze one earlier module | ↑ |
| RS | Right Shrink | $r \rightarrow r-1$ | Freeze the last trainable module | ↓ |
| RE | Right Expand | $r \rightarrow r+1$ | Unfreeze one later module | ↑ |
| PU | Push | $e \rightarrow e+1$ | Train one more module after exit | ↑ |
| PO | Pop | $e \rightarrow e-1$ | Stop one module earlier (pop exit) | ↓ |
| CON | Continue | — | Keep current state unchanged | ≈ |

## 4.2 ESTIMATORS FOR BENEFIT, MEMORY & TIME

To assess the impact of each candidate action, TriMS constructs and continuously updates three estimators during training: $\hat{V}$ for performance (i.e., loss) change, $\hat{M}$ for memory consumption, and $\hat{T}$ for time consumption. In this section, we present the design of these estimators, focusing on their modeling, initialization, and update processes.

**Modeling.** Operation 1 provides the evaluation methods for the three estimators (memory, time, and performance). For a given state $s_i = (l_i, s_i, e_i, n)$ and candidate action $a \in \mathcal{A}$, TriMS first computes the new state $s'$ after executing the action (line 1). Then, based on the new state $s'$, TriMS analyze

the memory and time costs in this state (line 3), with the following formulas:

$$\hat{M}(s) = M_B + \alpha_M \left( \sum_{j=1}^{e} M_P^{(j)} + \sum_{j=l}^{e} M_A^{(j)} + \sum_{j=l}^{r} M_G^{(j)} + \sum_{j=l}^{r} M_O^{(j)} \right),$$ (4)

$$\hat{T}(s) = T_B + \alpha_T \cdot \left( \sum_{j=1}^{e} T_{FP}^{(j)} + \sum_{j=l}^{e} T_{GP}^{(j)} + \sum_{j=l}^{r} T_{UP}^{(j)} \right).$$ (5)

Compared to Equation 1 and 2, we have added the weight coefficients $\alpha_M$ and $\alpha_T$ to better fit the actual memory and time consumption. For the performance benefit (i.e., the change in loss), TriMS analyzes the candidate actions using the activation gradients $\boldsymbol{g}$ obtained during training. TriMS calculates the performance benefit based on the module gradients affected by the action and the corresponding weights (lines 5-11). Different candidate actions affect different modules, and these effects are quantified using the corresponding sensitivity coefficients $\kappa_{\exp}$, $\kappa_{\text{shr}}$, and $\kappa_{\text{ext}}$, which represent the sensitivity coefficients for the expand, shrink, and exit operations, respectively. In this way, TriMS can estimate memory, time, and performance with minimal cost, providing efficient decision support for dynamic scheduling. The weights and parameters involved are initialized at the beginning of training and updated based on actual observations during training, improving the quality of the estimates.

---

**Operation 1: EstAction($\boldsymbol{s}_i, a$).**

---

**Input:** $\boldsymbol{s}_i = (l_i, r_i, e_i, n)$; $a \in \mathcal{A}$.
1   $\boldsymbol{s}' \leftarrow$ ApplyAction($\boldsymbol{s}_i, a$)
2   /* memory & time   */
3   $\hat{m}, \hat{t} \leftarrow \hat{M}(\boldsymbol{s}'), \hat{T}(\boldsymbol{s}')$
4   /* benefit $\hat{V}(\boldsymbol{s}, a)$   */
5   **if** $a = LE$ **then** $\hat{v} \leftarrow -\kappa_{\exp} \cdot \boldsymbol{g}^{(l_i - 1)}$
6   **elif** $a = RE$ **then** $\hat{v} \leftarrow -\kappa_{\exp} \cdot \boldsymbol{g}^{(r_i + 1)}$
7   **elif** $a = LS$ **then** $\hat{v} \leftarrow +\kappa_{\text{shr}} \cdot \boldsymbol{g}^{(l_i)}$
8   **elif** $a = RS$ **then** $\hat{v} \leftarrow +\kappa_{\text{shr}} \cdot \boldsymbol{g}^{(r_i)}$
9   **elif** $a = PU$ **then** $\hat{v} \leftarrow +\kappa_{\text{ext}} \cdot \boldsymbol{g}^{(e_i + 1)}$
10   **elif** $a = PO$ **then** $\hat{v} \leftarrow -\kappa_{\text{ext}} \cdot \boldsymbol{g}^{(e_i)}$
11   **else** $\hat{v} \leftarrow v_i - v_{i-1}$
**Output:** the estimated results $(\hat{m}, \hat{t}, \hat{v})$

---

**Operation 2: UpdateEst($\boldsymbol{s}_i, a_{i-1}$, OBS).**

---

**Input:** $\boldsymbol{s}_i = (l_i, r_i, e_i, n)$; $a_{i-1} \in \mathcal{A}$;
     OBS $= \{m_i, t_i, L_i, L_{i-1}\}$.
1   /* memory & time   */
2   $\alpha_M \leftarrow \text{EMA}(\alpha_M, \frac{m_i - M_B}{\hat{M}(\boldsymbol{s}_i) - M_B})$, $M_B \leftarrow m_i - \hat{M}(\boldsymbol{s}_i)$
3   $\alpha_T \leftarrow \text{EMA}(\alpha_T, \frac{t_i - T_B}{\hat{T}(\boldsymbol{s}_i) - T_B})$, $T_B \leftarrow t_i - \hat{T}(\boldsymbol{s}_i)$
4   /* benefit   */
5   **if** $a$ ***in*** $\{LE, RE\}$ **then** $\kappa_{\exp} \leftarrow \text{EMA}(\kappa_{\exp}, \frac{|L_i - L_{i-1}|}{|\sum_{j=r}^{l} \boldsymbol{g}^{(j)}|})$
6   **elif** $a$ ***in*** $\{LS, RS\}$ **then** $\kappa_{\text{shr}} \leftarrow \text{EMA}(\kappa_{\text{shr}}, \frac{|L_i - L_{i-1}|}{|\sum_{j=r}^{l} \boldsymbol{g}^{(j)}|})$
7   **elif** $a$ ***in*** $\{PU, PO\}$ **then** $\kappa_{\text{ext}} \leftarrow \text{EMA}(\kappa_{\text{ext}}, \frac{|L_i - L_{i-1}|}{|\sum_{j=e}^{l} \boldsymbol{g}^{(j)}|})$
8   **else** $\kappa_{\exp}, \kappa_{\text{shr}}, \kappa_{\text{ext}} \leftarrow \eta \cdot \kappa_{\exp}, \eta \cdot \kappa_{\text{shr}}, \eta \cdot \kappa_{\text{ext}}$
**Output:** the updated estimator $\hat{M}, \hat{T}, \hat{V}$

---

**Initialization.** At the beginning of training, TriMS first initializes the parameters for each estimator. **(i) Memory estimator.** TriMS computes the parameter memory $M_P^{(j)}$ based on the model weights and their data types, where $j \in [1, n]$. $M_G^{(j)}$ is set equal to $M_P^{(j)}$, while $M_O^{(j)}$ varies depending on the optimizer (e.g., for Adam optimizer, $M_O^{(j)} = 2M_P^{(j)}$). Additionally, the activation memory $M_A^{(j)}$ is initially set to $0.5M_P^{(j)}$, which serves as a rough estimate, and more refined estimates can be made based on batch size, sequence length, and other parameters. All coefficients are initialized as $\alpha_M = 1$, and after the first selection period, the difference between the peak memory $m_1$ measured during training and the estimated value $\hat{m}_1$ is used as $M_B$. **(ii) Time estimator.** For time consumption, TriMS initializes $T_{FP}, T_{GP}, T_{UP}$, and $T_B$ using the observed runtime from the first selection period. For modules without gradient propagation or parameter updates, their cost is approximated by $T_{FP}^{(j)}$. **(iii) Benefit estimator.** At the beginning of training, TriMS sets the initial state $\boldsymbol{s}_1 = (n, n, n, n)$ and the first action $a_1 = LE$. TriMS then initializes the performance benefit estimation parameters using the observed loss differences and boundary gradient strengths. Specifically, the sensitivity coefficient is initially computed based on the loss differences and gradients:

$$\kappa_{\exp} = \frac{|L_1 - L_0|}{|\boldsymbol{g}^{(n-1)}|}, \kappa_{\text{shr}} = (1 - \alpha)\kappa_{\exp}, \kappa_{\text{ext}} = (1 + \alpha)\kappa_{\exp}.$$ (6)

where $\alpha$ is a hyperparameter that controls the sensitivity difference between expansion, shrinkage, and exit operations. The shrinkage coefficient is larger because when the boundary shrinks, the

model typically limits the training region, directly impacting model performance. In contrast, the exit coefficient is smaller because early termination of modules usually results in a change in training depth, which has a more indirect effect on performance.

**Calibration.** During training, TriMS updates the parameters of its estimators using the observations collected at each selection period ($\tau$ iterations). The observations include the average loss $L_i$, peak memory $m_i$, total training time $t_i$, and activation gradients $\boldsymbol{g}^{(j)}, j \in [1, n]$. The activation gradient refers to the gradient strength of the loss with respect to each module's activation, which serves as a proxy for the module's training sensitivity. For efficiency in storage and backpropagation, gradients are computed only for modules in the range $[l-1, e]$, while the remaining modules are decayed with a factor $\eta$ (e.g., 0.99). The activation gradients are updated via exponential moving average (EMA):

$$\text{EMA}(g_{\text{prev}}, g_{\text{new}}) = \beta \cdot g_{\text{prev}} + (1 - \beta) \cdot g_{\text{new}}. \tag{7}$$

With these observations, TriMS updates the estimator parameters as illustrated in Operation 2. Specifically, the memory and time estimators update their coefficients $\alpha$ and base terms through EMA, while the performance benefit estimator updates depend on the previously executed action and selectively adjust the corresponding sensitivity parameters.

### 4.3 Action Selection and Scheduling Initialization

At each selection period, TriMS determines the next action to update the training state. The selection process is carried out in two steps.

**Step 1: Candidate filtering.** Given the current state $\boldsymbol{s}_i = (l_i, r_i, e_i, n_i)$, each candidate action $a \in \mathcal{A}$ is validated against structural and resource constraints. The structural constraint requires that the trainable block remain contiguous and confined to the non-exited modules, i.e., $1 \leq l \leq r \leq e \leq n$. The memory constraint ensures that the predicted memory consumption after applying the action, as estimated by the resource model, does not exceed the predefined budget $M_{\max}$. Any action that violates either constraint is filtered out.

**Step 2: Action evaluation and selection.** For the remaining candidate actions $a \in \mathcal{A}'$, TriMS computes a utility score based on both resource and performance estimations:

$$U(\boldsymbol{s}, a) = -\frac{\hat{v}}{L_i} - \lambda_M \frac{\hat{m} - m_i}{m_i} - \lambda_T \frac{\hat{t} - t_i}{t_i}, \tag{8}$$

where $(\hat{m}, \hat{t}, \hat{v})$ are estimates obtained from Operation 1, and $(m_i, t_i, L_i)$ are the values observed in the current selection period. The coefficients $\lambda_M$ and $\lambda_T$ control the penalties for memory and time consumption. The action with the highest $U(\boldsymbol{s}, a)$ is selected, yielding the next state $\boldsymbol{s}_{i+1}$.

**Scheduling initialization.** At the beginning of training, TriMS is initialized with a minimal configuration in which all modules are frozen except the last one, i.e., $\boldsymbol{s}_0 = (0, \ldots, 0, 1)$. In this state, the only feasible action is Left Expand (LE), which progressively enlarges the trainable region. This initialization provides a safe starting point while collecting essential resource and performance observations. After several initial steps, both the resource and benefit estimators are sufficiently warmed up, enabling TriMS to perform more flexible and accurate scheduling thereafter.

## 5 Experimental Study

To evaluate the effectiveness and rationality of our proposed method, we first report overall results across a diverse set of downstream tasks, followed by a series of ablation studies to examine the key design choices in TriMS. Our experiments cover 7 tasks (*mrpc*, *rte*, *qqp*, *qnli*, *mnli*, *cola*, and *stsb*) and three pre-trained models (i.e., RoBERTa-base, LLaMA-3.2-1B, and QWen2-0.5B), all of which can be fully fine-tuned on a single GPU, providing a practical setting for controlled comparison. We compared TriMS against several baselines, including full fine-tuning, head fine-tuning (configuring the head layer as trainable under the maximum memory constraint), LoRA (Hu et al., 2021), QLoRA (Dettmers et al., 2023), DoRA Liu et al. (2024a), and AdaLoRA (Zhang et al., 2023). All experiments are conducted on a Linux server equipped with an NVIDIA RTX A6000 GPU with 48GB memory. Additional implementation and training details are provided in the Appendix A.

## 5.1 OVERALL EXPERIMENT

We first compare different approaches under varying memory constraints. Specifically, we take the peak memory usage of full fine-tuning as a reference, and evaluate performance when the memory budget is restricted to $\{80\%, 60\%, 40\%\}$ of this value. For training, we set the batch size from $\{64, 32, 16\}$, using the largest value that can fit under full fine-tuning as the final configuration. Learning rates are swept over $\{1e-5, 1e-4, 1e-3\}$, and the best result is reported. For baselines, we tune memory-related hyperparameters, such as the number of trainable layers in head fine-tuning, or the rank size in LoRA (chosen from $\{2, 4, 8, 16, 32, 64, 128, 256, 512\}$). For TriMS, we run experiments with the default configuration. More detailed parameter settings and implementation details for all methods are provided in Appendix B.

**Overall performance.** Table 2 summarizes the average performance of different methods across tasks and memory budgets (measured by Pearson correlation for *stsb*, Matthews correlation for *cola*, and accuracy for others). The complete results are reported in Table 3 Appendix B. In the table, "–" denotes that the method cannot be applied to any task, while "∘" indicates applicability to only a subset of tasks. For methods that run on only part of the tasks, we assign the missing scores to the mean accuracy of other baselines on those tasks to compute the overall averages. The results show that almost all methods can operate under the 80% memory budget, with our method consistently ranking among the top two. However, when the memory budget becomes stricter, most baselines fail to adapt. QLoRA in particular struggles in many cases and becomes unusable under the 40% budget. In contrast, our method exhibits strong adaptability to strict memory constraints, maintaining competitive performance even under 60% and 40% budgets (e.g., with less than a 1.5% accuracy drop compared to full fine-tuning at 60% memory constraint).

Table 2: Average results (%) of baselines and our method on 7 tasks under different memory budgets.

| | 100% | 80% | | | | | | 60% | | | 40% | |
| | full | head | LoRA | QLoRA | DoRA | AdaLoRA | ours | head | QLoRA | ours | head | ours |
|---|---|---|---|---|---|---|---|---|---|---|---|---|
| RoBERTa | 82.97 | 82.62 | 82.85 | 79.63 | 80.69° | 80.57 | 82.84 | 79.79 | – | 81.19 | 75.98 | 79.47 |
| LLaMA | 83.82 | 84.02 | 83.26 | 82.05 | 83.43 | 81.23 | 84.31 | 80.56 | 78.34 | 82.36 | 74.42 | 76.76 |
| QWen | 80.50 | 80.71 | 77.76 | 79.36 | 80.89 | 80.67 | 81.61 | 78.79 | 78.57° | 79.68 | 71.90 | 76.37 |

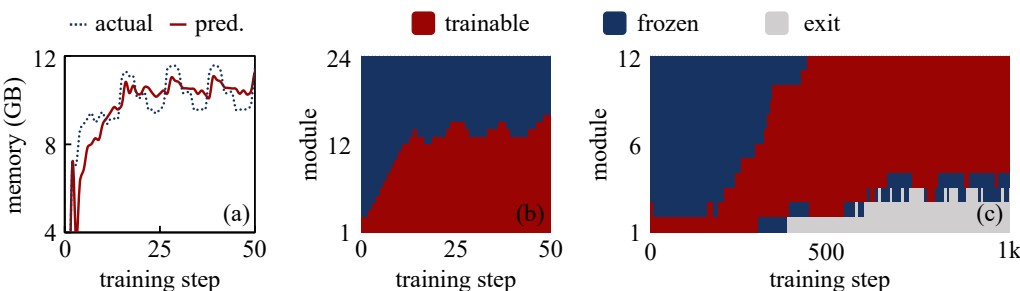

Figure 3: Memory monitoring and state transitions during training. (a) and (b) illustrate the memory usage and training state transitions of QWen on *mrpc* under an 80% memory constraint, while (c) shows the training state transitions of RoBERTa on *qqp*.

**Memory monitoring and state transitions.** To better understand the behavior of TriMS during training, we monitor both predicted and actual memory usage, together with the evolution of training states, as illustrated in Figure 3. In subplot (a), it can be observed that while the predicted values deviate from the actual measurements at the beginning of training, the update strategy quickly drives them closer. Although small discrepancies remain, the predictions are sufficient to ensure that training never exceeds the maximum memory budget. Subplots (b) and (c) further illustrate the transitions of training states under two different settings, showing that TriMS can automatically adapt its scheduling strategy according to task characteristics and resource constraints. Notably, on *qqp*, TriMS adopts an early-exit strategy, meaning that not all modules are trained, yet strong performance is still achieved: full fine-tuning reaches 91% accuracy, the best LoRA-based baseline achieves 88.49%, while our method obtains 90.55%.

## 5.2 INDEPENDENT EXPERIMENTS

To further validate the effectiveness of TriMS, we perform sensitivity analyses on its key hyper-parameters, as illustrated in Figure 4. These experiments provide insights into how different design choices affect both performance and resource efficiency. More detailed experimental settings are presented in Appendix C. The main findings are summarized as follows: **(i) Memory weight** $\lambda_M$ and **time weight** $\lambda_T$ in the scoring function (Equation 8). When evaluating candidate actions, TriMS balances performance gains from loss reduction with memory and time costs. Moderate weights yield better trade-offs between accuracy and resource efficiency, whereas excessively large weights overly restrict action selection and degrade performance. **(ii) Action selection period size** $\tau$. $\tau$ specifies the window size for estimating performance gains. A very small $\tau$ leads to noisy and inaccurate estimates, while an overly large $\tau$ reduces the frequency of action choices, limiting adaptability and lowering performance. **(iii) Sensitivity coefficient** $\alpha$ for $\kappa$ (Equation 6). $\alpha$ helps differentiate the impacts of expand ($\kappa_{\text{exp}}$), shrink ($\kappa_{\text{shr}}$), and exit ($\kappa_{\text{ext}}$) actions. A moderate value improves action discrimination, while an excessively large $\alpha$ has the opposite effect, unnecessarily restricting the selection of certain actions. **(iv) EMA weight** $\beta$ (Equation 7). $\beta$ smooths updates of activation gradients and estimator parameters, mitigating short-term fluctuations. An appropriate $\beta$ enhances stability and performance, whereas a very small $\beta$ makes updates overly dependent on recent results, increasing volatility and harming final performance.

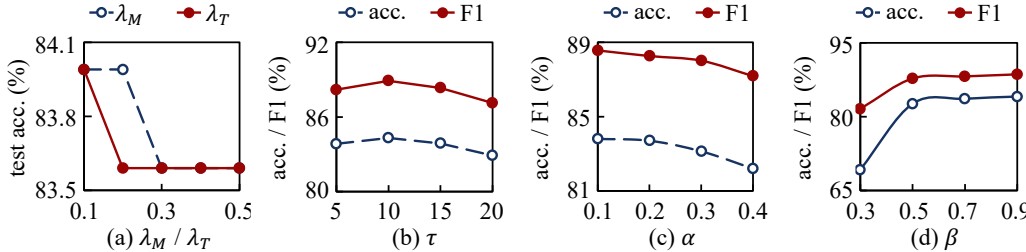

Figure 4: Hyperparameter sensitivity of TriMS on QWen with the *mrpc* task under the 80% memory budget: (a) memory weight $\lambda_M$ and time weight $\lambda_T$ in the scoring function; (b) action selection interval $\tau$; (c) sensitivity coefficient $\alpha$ for $\kappa$; (d) EMA weight $\beta$.

## 6 CONCLUSION

GPU memory consumption remains a major bottleneck for the practical fine-tuning of large language models. Existing memory-efficient methods, due to the lack of explicit modeling of memory cost and consideration of dynamic adjustment, are often limited in both efficiency and adaptability. To address this challenge, we propose Three-state Module Scheduling (TriMS), a dynamic fine-tuning framework that integrates three-state modeling (i.e., trainable, frozen, and early exit) with a performance–cost estimator to adaptively schedule module states throughout training. By explicitly modeling memory usage and characterizing training configurations, TriMS provides a principled way to reason about resource consumption. Moreover, by dynamically adjusting to training dynamics, it enhances controllability and robustness while maintaining high efficiency. This design enables TriMS to operate reliably even under strict memory budgets, bridging the gap between practical resource constraints and effective fine-tuning. Extensive experiments demonstrate that TriMS achieves robust performance across diverse tasks and models, matching or even surpassing the best baselines. Even under stricter memory budgets (e.g., 60% of the peak memory in full fine-tuning), where other methods cannot fit into memory, TriMS remains effective and achieves accuracy close to full fine-tuning (less than a 1.5% accuracy drop).

**Limitations and future work** In this work, we constrain trainable modules to be consecutive in order to simplify both action design and estimator construction. While more flexible configurations might yield better results, they would also introduce additional complexity, which we plan to investigate in future work. Moreover, as model size continues to grow and parameter storage requires multi-GPU settings, extending TriMS to support efficient cross-device scheduling remains an important direction for future exploration.

## REPRODUCIBILITY STATEMENT

All models, datasets, and baseline methods used in this work are publicly available. Detailed descriptions of experimental settings, hyperparameter configurations, and training procedures are provided in the Experiments section and further elaborated in the Appendix. To ensure full reproducibility, we will also release the complete implementation and scripts for all experiments.

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

## A    DATASETS, MODELS, AND EXPERIMENTAL SETTINGS

In this section, we provide additional details regarding the datasets, models, and experimental settings in our experiments.

**Datasets.** We used 6 NLP tasks in the experiments, including *mrpc*, *qnli*, *mnli*, *qqp*, *cola*, and *stsb*. All the datasets were loaded from HuggingFace Wolf et al. (2020).

- *mrpc* (accuracy): Microsoft Research Paraphrase Corpus, which consists of 5.8k sentence pairs that were automatically extracted from online news sources. The sentence pairs have been annotated by human raters to indicate whether the sentences within each pair are semantically equivalent.

- *rte* (accuracy): the recognizing textual entailment datasets come from a series of annual textual entailment challenges. Examples are constructed based on news and Wikipedia text. The authors of the benchmark convert all datasets to a two-class split, where for three-class datasets they collapse neutral and contradiction into not entailment, for consistency.

- *qnli* (accuracy): Stanford Question Answering Dataset, which is a question-answering dataset consisting of question-paragraph pairs, where one of the sentences in the paragraph (drawn from Wikipedia) contains the answer to the corresponding question (written by an annotator).

- *mnli* (accuracy): Multi-Genre Natural Language Inference Corpus, a crowdsourced collection of sentence pairs with textual entailment annotations.

- *qqp* (accuracy): the Quora Question Pairs dataset, which consists of over 400,000 pairs of questions. Each question pair is annotated with a binary value indicating whether the two questions are paraphrases of each other.

- *cola* (matthews correlation): Corpus of Linguistic Acceptability consists of English acceptability judgments drawn from books and journal articles on linguistic theory.

- *stsb* (pearson): Semantic Textual Similarity Benchmark, a collection of sentence pairs drawn from news headlines, video and image captions, and natural language inference data. Each pair is human-annotated with a similarity score from 1 to 5.

**Models.**    For the pre-trained models, we used three pre-trained NLP models (i.e., RoBERTa (Camacho-collados et al., 2022), LLaMA-3.2-1B (Touvron et al., 2023), and QWen2-0.5B (Yang et al., 2024)) from HuggingFace (Wolf et al., 2020). We deliberately chose models that can be fully fine-tuned on a single GPU, ensuring that all compared methods are evaluated under the same hardware conditions and allowing for a fair comparison of memory efficiency and training effectiveness. Since full fine-tuning larger-scale models typically requires multi-GPU distributed training, they introduce additional communication and parallelization overheads and make memory measurement and comparison less consistent, which could obscure the advantages of our method in the single-GPU setting.

**Baselines.** For baseline comparisons, we adopted several representative fine-tuning methods: full fine-tuning, head fine-tuning, LoRA (Hu et al., 2021), QLoRA (Dettmers et al., 2023), DoRA Liu et al. (2024a), and AdaLoRA (Zhang et al., 2023).

**Setup.** In our experiments, we searched over batch sizes $\{16, 32, 64\}$ and token lengths $\{128, 256\}$, and selected the largest configuration that allowed full fine-tuning to run on a single GPU. For the baselines, we trained each method using the maximum feasible hyperparameter values that fit into memory: for head fine-tuning, this corresponds to the maximum number of trainable modules that can be accommodated, and for LoRA-based methods, the largest rank within $\{2, 4, 8, 16, 32, 64, 128, 256, 512\}$. In addition, we tuned the learning rate over $\{1e\text{-}3, 1e\text{-}4, 1e\text{-}5\}$ and the number of training epochs over $\{2, 5, 10\}$, and optionally enabled early stopping (terminating training when the loss decreased by less than $10^{-4}$ for 10 consecutive evaluations). The best result from these configurations is reported for each baseline. For TriMS, we trained with fixed hyperparameters: $\alpha = 0.1$, $\beta = 0.9$, $\tau = 10$, and $\lambda_M = \lambda_T = 0.1$.

Table 3: Average results (%) of baselines and our method on 7 tasks under different memory budgets.

| Model | Ratio (%) | Method | mrpc | rte | qqp | qnli | mnli | cola | stsb |
|---|---|---|---|---|---|---|---|---|---|
| RoBERTa | 100% | full | 87.30 | 75.09 | 91.00 | 92.39 | 86.97 | 58.30 | 89.74 |
| | 80% | head | 86.03 | 75.81 | 89.97 | 91.67 | 86.69 | 59.44 | 88.70 |
| | | LoRA | 85.45 | 70.40 | 88.49 | 90.88 | 86.32 | 69.49 | 88.95 |
| | | QLoRA | 84.49 | 73.82 | 86.82 | 90.96 | 83.45 | 50.19 | 87.66 |
| | | DoRA | – | – | 85.96 | – | 82.78 | 58.43 | 88.89 |
| | | AdaLoRA | 86.26 | 70.04 | 88.22 | 90.24 | 85.49 | 54.85 | 88.92 |
| | | ours | 86.91 | 75.52 | 90.55 | 91.89 | 86.30 | 59.44 | 89.25 |
| | 60% | head | 84.06 | 68.95 | 88.61 | 89.64 | 83.25 | 57.02 | 86.97 |
| | | QLoRA | – | – | – | – | – | – | – |
| | | ours | 85.89 | 71.32 | 88.51 | 90.23 | 85.91 | 58.71 | 87.79 |
| | 40% | head | 66.49 | 67.15 | 86.70 | 86.33 | 85.81 | 56.00 | 83.35 |
| | | ours | 79.14 | 71.02 | 87.95 | 88.81 | 86.21 | 57.93 | 85.23 |
| LLaMA | 100% | full | 83.30 | 83.39 | 91.06 | 93.01 | 86.45 | 59.78 | 89.78 |
| | 80% | head | 84.41 | 84.28 | 90.08 | 92.67 | 86.12 | 61.17 | 89.44 |
| | | LoRA | 83.71 | 79.78 | 90.16 | 92.79 | 87.13 | 60.21 | 89.03 |
| | | QLoRA | 82.88 | 80.37 | 88.63 | 91.27 | 86.65 | 59.65 | 84.90 |
| | | DoRA | 83.19 | 80.87 | 90.76 | 93.79 | 86.14 | 60.13 | 89.11 |
| | | AdaLoRA | 79.64 | 78.06 | 89.72 | 92.24 | 85.92 | 55.83 | 87.21 |
| | | ours | 84.39 | 84.31 | 90.56 | 92.81 | 86.71 | 61.59 | 89.78 |
| | 60% | head | 77.68 | 79.42 | 88.03 | 91.32 | 85.39 | 59.04 | 83.04 |
| | | QLoRA | 82.88 | 61.37 | 88.63 | 91.27 | 86.65 | 53.65 | 83.90 |
| | | ours | 83.28 | 82.36 | 89.10 | 91.89 | 85.57 | 59.93 | 84.41 |
| | 40% | head | 74.72 | 67.15 | 87.45 | 87.13 | 83.13 | 49.77 | 71.58 |
| | | ours | 77.15 | 75.21 | 87.75 | 87.81 | 84.93 | 50.12 | 74.32 |
| QWen | 100% | full | 84.06 | 76.17 | 90.42 | 89.40 | 84.60 | 53.18 | 85.67 |
| | 80% | head | 83.07 | 76.90 | 90.98 | 89.20 | 86.30 | 54.44 | 84.09 |
| | | LoRA | 83.94 | 75.45 | 83.88 | 81.55 | 78.10 | 55.22 | 86.12 |
| | | QLoRA | 82.20 | 73.65 | 88.36 | 88.94 | 83.96 | 51.37 | 87.06 |
| | | DoRA | 84.93 | 77.98 | 88.39 | 89.20 | 84.73 | 54.01 | 86.99 |
| | | AdaLoRA | 83.88 | 74.73 | 89.51 | 89.93 | 85.30 | 56.11 | 85.25 |
| | | ours | 83.99 | 77.86 | 91.03 | 89.71 | 86.45 | 55.32 | 86.92 |
| | 60% | head | 79.54 | 74.01 | 90.23 | 89.18 | 85.35 | 50.62 | 82.63 |
| | | QLoRA | 82.20 | – | 88.10 | 88.13 | 83.96 | – | 82.97 |
| | | ours | 82.49 | 75.51 | 90.57 | 89.20 | 86.13 | 51.02 | 82.87 |
| | 40% | head | 70.55 | 64.62 | 89.06 | 84.84 | 80.47 | 44.77 | 68.98 |
| | | ours | 77.31 | 71.03 | 90.35 | 86.83 | 82.31 | 48.65 | 78.09 |

# B   MORE DETAILS ABOUT THE OVERALL EXPERIMENT

Table 2 reports the average results of different methods across tasks under varying memory budgets, while Table 3 presents the detailed per-task results. From the detailed results, we can observe that TriMS consistently achieves strong performance across diverse tasks and models. Notably, its ad-

vantage becomes more pronounced under stricter memory budgets, where other baselines cannot fit into memory, whereas TriMS remains effective and competitive.

## C  MORE DETAILS ABOUT THE INDEPENDENT EXPERIMENT

In the independent experiments, we further examined the impact of different hyperparameters on the performance of TriMS. These experiments were conducted on the *mrpc* task with QWen under an 80% memory budget. All default hyperparameter values were kept consistent with those used in the overall experiments to ensure comparability.

## D  LLM USAGE STATEMENT

Large language models (LLMs) were only used in this work to assist with writing, including language polishing and grammar checking.

