# OpenReview forum: "Train, Freeze or Exit: Dynamic Module-wise Fine-Tuning under Memory Constraints"
_ICLR.cc/2026/Conference — ICLR 2026 Conference Withdrawn Submission_

### Official Review · Reviewer_Rfnb · 2025-10-21

**Soundness:** 2
**Presentation:** 2
**Contribution:** 2
**Rating:** 4
**Confidence:** 4

**Summary:**

This paper proposes TriMS, a dynamic fine-tuning framework. By introducing a three-state module formulation (trainable/frozen/early-exit) with continuity constraints, TriMS enables explicit resource modeling and transforms dynamic scheduling into actionable candidate selections.

**Strengths:**

> s1: Figures are of high quality.

> s2: Introduces a unified state representation (trainable, frozen, early exit) for modules, enabling explicit and quantitative estimation of memory and time costs for different training configurations.

> s3: Dynamic scheduling framework with continuity constraints and seven actionable candidates.

**Weaknesses:**

> w1: Lines 133-157. Please clarify the distinction between "trainable (1)" and "early exit (-1)". For example, when the state vector s=[1,1,-1], **does this mean the model's output is not taken from the final layer of the base model? If so, could this introduce bias during fine-tuning?** If not, the current presentation seems ambiguous and requires detailed explanation.

> w2: Lines 204-206: "we incorporate a contiguity constraint that requires all trainable modules to form a continuous block".  Please explain **which state (trainable 1/frozen 0/early-exit -1) corresponds to a continuous value in state vector s**, and how this should be interpreted.

> w3: Lines 278-281: "we have added the weight coefficients αM and αT to better fit the actual memory and time consumption." Absence of discussion regarding relevant weighting coefficients, such as how to select appropriate/optimal parameter configurations.

> w4: Lines 345-347: "each candidate action a ∈ A is validated against structural and resource constraints." How was the action set obtained? The main text lacks specific explanation on this matter.

> w5: For Section 5.1 (Tables 2 and 3).  Interestingly, in the experimental results presented by the authors, the fine-tuning performance of the llama-3.2-1B model on task like mnli appears inferior to that of the roberta model. **This contradicts the experimental observations** reported in works such as LoRA+ [1], which is rather perplexing.
>
> [1] LoRA+: Efficient Low Rank Adaptation of Large Models. ICML 2024.

> w6: In Table 2, the authors present the average results (%) of baselines and their method across 7 tasks. However, tasks like mrpc and rte have significantly fewer data instances compared to tasks like mnli, **making averaging across all tasks a suboptimal approach**.

**Questions:**

See Weaknesses. I would reconsider my score if these concerns are adequately addressed.

---

### Official Review · Reviewer_SVCW · 2025-10-25

**Soundness:** 3
**Presentation:** 2
**Contribution:** 2
**Rating:** 6
**Confidence:** 4

**Summary:**

The paper studies fine-tuning large (but single-GPU-fit) language models under strict memory budgets. It proposes Three-State Module Scheduling, which encodes each Transformer block as trainable (1), frozen (0), or early-exit (-1). Under a contiguity constraint, the scheduler chooses among seven actions (left/right expand or shrink, push/pop the exit depth, or continue). TriMS maintains lightweight estimators for memory, time, and performance benefit (the latter derived from EMA-smoothed activation gradients at module boundaries). At fixed intervals, it filters actions by budget feasibility and selects the highest utility per a benefit–cost score.
Experiments on 7 GLUE-style tasks (MRPC, RTE, QQP, QNLI, MNLI, CoLA, STS-B) and three models (RoBERTa-base, LLaMA-3.2-1B, Qwen2-0.5B) show that, relative to strong PEFT baselines (LoRA/QLoRA/DoRA/AdaLoRA) and head-only tuning, TriMS is competitive at 80% of full-FT peak memory and degrades more gracefully at 60%/40% budgets; in some cases, it matches full FT within ~1–1.5% accuracy at 60% memory. The paper also provides sensitivity studies for the score weights, selection period, gradient EMA, and benefit coefficients.

**Strengths:**

The three-state encoding with a contiguity constraint yields a compact, controllable action space with explicit feasibility checks. Modeling memory/time (and calibrating via EMA) reduces trial-and-error and provides a principled way to avoid OOM while adapting the configuration. The scheduler captures stage-dependent needs of different depths (expand/shrink/push/pop), which classic fixed-rank LoRA cannot. At 60% memory, TriMS remains within ~1–1.5% of full FT in several settings and outperforms common PEFT baselines that degrade or fail at 40–60%.

**Weaknesses:**

The paper positions itself against AFLoRA, ALaST, DLFT, etc.. Still, it should more sharply differentiate the benefit estimator and early-exit depth control from prior adaptive layer/rank selection beyond “heuristic vs. predictive” language. A stronger comparative analysis (incl. more recent 2024–2025 dynamic PEFT variants) would clarify novelty.

The performance estimator relies on boundary activation gradients and several coefficients (κ_exp/κ_shr/κ_ext, α, β). Theoretical justification is limited; oscillation/instability risks are not deeply analyzed. Provide evidence that the utility score is robust across seeds/tasks, and discuss failure modes.

The setting is realistic for budget users, but broader evidence (multi-GPU, 7B-class models, instruction tuning, or generative tasks) would increase significance and external validity.

**Questions:**

Which of the following were enabled/tuned for baselines at low budgets: gradient checkpointing, optimizer state offloading, ZeRO-style sharding, parameter-offloading to CPU/NVMe, sequence-length scaling, micro-batching? Please provide a table of per-baseline memory knobs and achieved peak memory.

Why boundary activation gradients (vs. layerwise loss influence or Fisher/sensitivity measures)? Have you compared κ learned by regression on held-out mini-batches vs. fixed EMA?

Any preliminary results on a 7B-class model (even a short run) or on multi-GPU with activation/optimizer sharding?

---

### Official Review · Reviewer_w6hR · 2025-10-27

**Soundness:** 2
**Presentation:** 3
**Contribution:** 3
**Rating:** 4
**Confidence:** 4

**Summary:**

The paper introduces ​​Three-State Module Scheduling (TriMS)​​, a dynamic fine-tuning framework designed for strict memory constraints. Its core innovation lies in a ​​three-state encoding​​ combined with a ​​contiguity constraint​​ that simplifies the action space to seven discrete operations. During training, TriMS operates in a closed loop: it ​​monitors​​ performance metrics and resource usage, ​​updates​​ predictive estimators for memory, time, and performance benefit, and ​​selects​​ the optimal action based on a utility score that balances estimated performance gain against resource cost. This data-driven approach enables ​​adaptive module scheduling​​, allowing TriMS to efficiently maximize model performance under a fixed memory budget by dynamically adjusting which parts of the model are trained.

**Strengths:**

1. This paper is well-written and easy to understand the core contributions and method.

2. The TriMS is innovative and it is good to incorporate the `Reinforcement Learning Principle` into the design of the method.

**Weaknesses:**

1. I think the overall method of this paper is innovative, as it draws inspiration from PPO methods in reinforcement learning, although the authors may not have explicitly emphasized this, the approach itself embodies a similar philosophy. However, the experimental section of the paper does feel somewhat outdated. I can understand that hardware limitations may have prevented the authors from training larger-scale models, but the tasks chosen for evaluation do not align well with the current progress in LLM research. I would have preferred to see experiments conducted on more contemporary tasks, such as multi-turn dialogue data, mathematical reasoning tasks, or code generation tasks. Relying solely on traditional NLP tasks no longer seems sufficiently convincing.

2. Regarding the shortcomings of the experiment, a large number of baselines already exist in this field, such as the DLFT method mentioned by the authors themselves in the Related Work section, as well as approaches like LISA and HFT, which freeze certain modules and only fine-tune others. I believe that comparing solely with some classic methods of the LoRA series in the experiment is insufficient to demonstrate the effectiveness of the proposed method. I hope the authors can include more recent works as baselines (excluding concurrent works from the past three months).

3. The paper lacks a detailed ablation study to verify the effectiveness of each component.

Although I find the method proposed in the paper quite interesting and its motivation well-justified, I believe the richness of the experimental section is equally important. In my view, the current version of the paper lacks many critical experiments. Therefore, I would be willing to raise my score if the authors supplement the relevant experiments.

**Questions:**

Looking at the content of Figure 3, does the approach in this paper involve first training only the initial or final module individually while freezing all others, and then, based on the set actions, allowing the trainable portion to progressively extend toward the other end? For example, if initially only the first layer is trainable while the rest are frozen, and the action is constrained to move one step at a time, does this imply that the expansion of trainable layers progressively advances toward the deeper layers?

---

### Official Review · Reviewer_c2G1 · 2025-11-01

**Soundness:** 2
**Presentation:** 2
**Contribution:** 2
**Rating:** 2
**Confidence:** 4

**Summary:**

This paper addresses the problem of efficient fine-tuning of large language models under memory constraints. Existing parameter-efficient methods fail to explicitly model memory cost, and lack dynamic adaptability. This paper proposes TriMS — a fine-tuning framework that explicitly models memory and time costs for each module and dynamically schedules their states among trainable, frozen, and early-exited options.
At each update step, TriMS estimates performance and makes optimal action selection based on estimation and constraint. Such methods can dynamically change the trainable modules and achieve maximized memory efficiency under a certain budget.
Experiments on multiple LLMs show promising results under certain memory budget constraints.
The main contributions are:
(1) A three-state representation (trainable, frozen, early-exited) with structural constraints, enabling dynamic adaptability of fine-tuning size under varying memory budgets; and
(2) An optimizable estimator of performance gain, which guides action selection to achieve near-optimal training configurations within the given resource limit.

**Strengths:**

**Originality.**
The paper introduces a three-state representation (trainable, frozen, early-exited) that forms a sound framework for dynamic fine-tuning under memory constraints.
It also addresses two previously unresolved issues: the absence of explicit modeling of memory cost, and the lack of an adaptive mechanism balancing performance and resource use.

**Quality.**
The overall framework is technically coherent and well-structured, effectively linking performance estimation with resource-aware scheduling.

**Significance.**
The proposed framework addresses an important and timely challenge in large-model fine-tuning — maintaining accuracy under strict hardware constraints. By formulating fine-tuning as a dynamic scheduling problem, TriMS provides a potentially general paradigm for resource-aware adaptation of LLMs.

**Weaknesses:**

**Representation and clarity.**
The paper contains numerous presentation errors and inconsistencies.
Symbols and variables are frequently redefined or misused:
‘s’ appears as a per-module state (Line 143), a per-step state (Line 195), and an index of trainable modules (Line 169).
action symbol ‘a’ replaced by \alpha at line 216.
Equations 4 and 5 define \alpha as weight coefficients, while Line 323 redefines it as a sensitivity hyperparameter.
Operation 1 introduces \hat m, \hat t, and \hat v without definition.
Figure 2 lacks proper legends: the red dashed line linking the estimator and action is undefined, and two triangle symbols (solid vs hollow) are used without explanation.
Figure 2 illustration is very ambiguous. The performance variable L is undefined, and its connection to V (the evaluation metric) is unclear.

**Lack of Motivation.**
Key components—including the use of activation gradients for performance estimation, the initialization rule M_A=0.5M_P, and the EMA update procedure—lack motivation reasoning and ablation study.

**Experimental evaluation.**
The TriMS framework appears to introduce substantial additional computation during fine-tuning compared to other PEFT methods. A direct comparison of fine-tuning efficiency and training time is necessary to demonstrate the framework’s practical feasibility. Including an additional analysis on how fine-tuning efficiency scales with model size would further strengthen the claim that TriMS remains controllable for larger models.
The “independent experiments” section performs only sensitivity analysis and lacks rigorous ablation studies. Without explicit ablations removing key components, it is unclear whether the proposed mechanisms are individually effective or simply correlated with other design factors.

**Citation and context.**
Several models and downstream tasks are stated without in-text citation.

**Questions:**

In Figure 1, do the early-exit modules each have an independent task head, or do they share a single common head?


In Section 3.3, V(\cdot) is evaluated by accuracy or F1 score, but in Section 4.2, \hat{V} is defined in terms of loss. Is this a typographical error or an intentional design choice?


What is the motivation for using activation gradients to estimate performance improvement?


Similarly, what is the rationale for introducing the EMA update mechanism? Figure 3a shows that predicted memory values become smoother than actual measurements in later training steps. Is this smoothing effect caused by EMA? If EMA were removed, would the predictions track reality more closely, and would this improve performance?


The paper mentions that when some downstream tasks cannot be executed, average accuracy is used instead. Is this substitution statistically sound, or could it introduce evaluation bias?


In Table 2, the “Head” baseline appears to refer to head fine-tuning, whose actual memory usage is much lower than full fine-tuning. How can such a baseline be fairly compared under the same nominal memory budgets?


Figures 3b and 3c are described as showing results under “two different settings,” but these settings are not specified. What exactly differentiates the two configurations?

---

### Note · Authors · 2025-12-04

**Comment:**

Dear AC and Reviewers,

We sincerely thank all reviewers and the area chair for their detailed and constructive feedback. After careful consideration, we have decided to withdraw the submission. We have made further revisions addressing the comments regarding presentation clarity, the motivation of key components, and the completeness of experiments and ablations. However, due to limitations in time and computational resources, we are currently unable to conduct additional experiments on larger models or datasets. Therefore, we would like to reserve more time to further improve the work. We sincerely appreciate the reviewers’ and AC’s time and thoughtful feedback. Thank you again for the valuable comments, which will help guide the continued refinement of this work.

Authors of Submission #3549

**Withdrawal Confirmation:**

I have read and agree with the venue's withdrawal policy on behalf of myself and my co-authors.